# Cluster Analysis Identifies Distinct Patterns of T-Cell and Humoral Immune Responses Evolution Following a Third Dose of SARS-CoV-2 Vaccine in People Living with HIV

**DOI:** 10.3390/v15071435

**Published:** 2023-06-26

**Authors:** Majdouline El Moussaoui, Salomé Desmecht, Nicolas Lambert, Nathalie Maes, Joachim Braghini, Nicole Marechal, Céline Quintana, Karine Briquet, Stéphanie Gofflot, Françoise Toussaint, Marie-Pierre Hayette, Pieter Vermeersch, Laurence Lutteri, Céline Grégoire, Yves Beguin, Souad Rahmouni, Michel Moutschen, Daniel Desmecht, Gilles Darcis

**Affiliations:** 1Department of Infectious Diseases and General Internal Medicine, University Hospital of Liège, 4000 Liège, Belgium; 2Laboratory of Animal Genomics, GIGA-Medical Genomics, GIGA-Institute, University of Liège, 4000 Liège, Belgium; 3Department of Neurology, University Hospital of Liège, 4000 Liège, Belgium; 4Biostatistics and Research Method Center (B-STAT), University Hospital of Liège, 4000 Liège, Belgium; 5Department of Biothèque Hospitalo-Universitaire de Liège (BHUL), University Hospital of Liège, 4000 Liège, Belgium; 6Department of Clinical Microbiology, University Hospital of Liège, 4000 Liège, Belgium; 7Department of Laboratory Medicine, University Hospital of Leuven, 3000 Leuven, Belgium; 8Department of Clinical Chemistry, University Hospital of Liège, 4000 Liège, Belgium; 9Department of Haematology, University Hospital of Liège, University of Liège, 4000 Liège, Belgium; 10Department of Animal Pathology, Fundamental and Applied Research for Animals & Health, University of Liège, 4000 Liège, Belgium

**Keywords:** SARS-CoV-2 mRNA vaccine, HIV, antibodies, humoral, cellular, immune response, neutralisation, third dose, Omicron, people living with HIV

## Abstract

(1) Background: Many vaccines require higher, additional doses or adjuvants to provide adequate protection for people living with HIV (PLWH). Despite their potential risk of severe coronavirus disease 2019, immunological data remain sparse, and a clear consensus for the best booster strategy is lacking. (2) Methods: Using the data obtained from our previous study assessing prospective T-cell and humoral immune responses before and after administration of a third dose of SARS-CoV-2 vaccine, we assessed the correlations between immune parameters reflecting humoral and cellular immune responses. We further aimed at identifying distinct clusters of patients with similar patterns of immune response evolution to determine how these relate to demographic and clinical factors. (3) Results: Among 80 PLWH and 51 healthcare workers (HCWs) enrolled in the study, cluster analysis identified four distinct patterns of evolution characterised by specific immune patterns and clinical factors. We observed that immune responses appeared to be less robust in cluster A, whose individuals were mostly PLWH who had never been infected with SARS-CoV-2. Cluster C, whose individuals showed a particularly drastic increase in markers of humoral immune response following the third dose of vaccine, was mainly composed of female participants who experienced SARS-CoV-2. Regarding the correlation study, although we observed a strong positive correlation between markers mirroring humoral immune response, markers of T-cell response following vaccination correlated only in a lesser extent with markers of humoral immunity. This suggests that neutralising antibody titers alone are not always a reliable reflection of the magnitude of the whole immune response. (4) Conclusions: Our findings show heterogeneity in immune responses among SARS-CoV-2 vaccinated PLWH. Specific subgroups could therefore benefit from distinct immunization strategies. Prior or breakthrough natural infection enhances the activity of vaccines and must be taken into account for informing global vaccine strategies among PLWH, even those with a viro-immunologically controlled infection.

## 1. Introduction

A wide range of highly efficient vaccines against the severe acute respiratory syndrome coronavirus 2 (SARS-CoV-2) has been developed with unprecedented speed [1]. Several studies [2,3,4] have characterised T-cell and humoral immune responses against SARS-CoV-2, demonstrating that most people generate both virus-specific antibodies and T-cells after vaccination. Vaccines induce virus spike protein-specific antibodies, and their neutralising capacity’s magnitude positively correlates with disease severity [5]. Besides humoral immune responses, accumulating data suggest that T-cell immunity plays an important role in vaccine protection against severe COVID-19 disease, particularly against viral variants that partially escape from recognition by neutralising antibodies like the Omicron variant [3,6,7]. Such vaccine efficacy is not reached in all individuals. Due to their immunocompromised state, people living with HIV (PLWH) groups were underrepresented in the initial phase III vaccine efficacy trials and deserve special attention when evaluating their vaccine responses [8,9]. In a recently published study, we prospectively characterised T-cell and humoral immune responses following a third dose of SARS-CoV-2 vaccine in a population-based cohort of 80 PLWH followed up at the University Hospital of Liège (Belgium) and in 51 HIV-negative healthcare workers (HCWs), demonstrating that the vaccine induced robust T-cell and humoral immune responses against SARS-CoV-2 in almost all participants [10]. We further contrasted our results according to participants’ prior SARS-CoV-2 infection based on anti-nucleocapsid Ig and a questionnaire. Humoral immune response assessed in terms of anti-spike (anti-S) IgG was similar between PLWH and HCWs, both before and after the third dose, regardless of the SARS-CoV-2 infection history. While the proportion of detectable neutralising antibodies and titers against both wild type (Wuhan-like) and Omicron strains (BA.1/B.1.1.529) increased significantly following the administration of the third dose, neutralising antibody titers (nAbTs) against Omicron remained eight-fold lower compared to those against wild type, which may reflect less effective protection against this variant.

Although SARS-CoV-2 specific IFN-ɣ production increased after the third dose, it remained significantly lower among SARS-CoV-2 naïve PLWH compared to HCWs. In contrast, hybrid immunity, emerging from both infection-induced and vaccine, conferred similar T-cell immune responses following the administration of the third dose between PLWH and HIV-negative individuals, suggesting a potential protective advantage of hybrid immunity in PLWH. Interestingly, subgroup analyses according to CD4^+^ T cell count or CD4^+^/CD8^+^ T cell ratio did not reveal any significant difference between immune responses of PLWH. Therefore, our data raise concerns about the vaccine’s ability to induce a protective T-cell immune response among PLWH with no history of SARS-CoV-2 infection. Most individuals generate virus-specific T-cell responses, but these are heterogeneous and may provide various protection against severe COVID-19. Using the data obtained from our previous analyses, we explored the correlations within and between vaccine-induced T-cell and humoral immune responses before and after the administration of the third vaccine dose. Based on the co-evolution of T-cell and humoral immune responses over time, we further aimed to identify distinct clusters that independently correspond to specific patterns of immune responses and to determine how these relate to demographic and clinical factors.

## 2. Materials and Methods

### 2.1. Study Design and Participants

In this prospective, two-arms, non-randomised, monocentric study, we enrolled a cohort of HIV-infected individuals under routine follow-up at the University Hospital of Liège, Belgium, and a cohort of HIV-negative control individuals composed of HCWs from the same institution. All participants were 18 years of age or above and had received two doses of vaccine against SARS-CoV-2 (either BNT162b2, mRNA-1273, or ChAdOx1-S).

### 2.2. Procedures

We collected baseline characteristics of PLWH from electronic medical registry. Equivalent data were collected from HCWs through a questionnaire, completed between 22 February and 5 March 2021, during their participation in another prospective study evaluating the seroprevalence of anti-SARS-CoV-2 IgG antibodies [11]. History of confirmed SARS-CoV-2 infection was evaluated through quantification of anti-SARS-CoV-2 anti-nucleocapsid total antibodies using Elecsys^®^ anti-SARS-CoV-2 assay on a Cobas e801 module analyser (Roche Diagnostics, Basel, Switzerland) and a questionnaire completed at each sampling time point.

The booster dose, either BNT162b2 or mRNA-1273, was administered through Belgium’s vaccination campaign independently of the study protocol. Peripheral blood from each patient was sampled before (T0) and two to eight weeks after the third dose of vaccine (T1). The samples at T0 were collected between August 2021 and September 2021 for HCWs and between December 2021 and January 2022 for PLWH. The samples at T1 were collected between January and February 2022 for both groups at T1. Biological analyses at each sampling timepoint included detection and quantification of anti-trimeric spike protein specific IgG antibodies (anti-S IgG) using the Liaison^®^ SARS-CoV-2 TrimericS IgG chemiluminescent immunoassay on the Liaison XL analyser (Saluggia, Italy), 50% neutralising antibody titers (NT_50_) against the wild type (WT) (Wuhan-like) and Omicron (BA.1/B.1.1.529) strains, and SARS-CoV-2-specific interferon-gamma (IFN-γ) release using the QuantiFERON SARS-CoV-2 assay (Saluggia, Italy, DiaSorin) which contains two different pools (Ag1 and Ag2) of spike-embedded peptides.

#### 2.2.1. Quantification of Total SARS-CoV-2 Anti-Nucleocapsid Total Antibodies

Total antibodies against SARS-CoV-2 nucleocapsid were measured using the electrochemiluminescent immunoassay (ECLIA) Elecsys^®^ anti-SARS-CoV-2 assay on Roche Cobas e801 (Roche Diagnostics, Basel, Switzerland) according to the manufacturer’s instructions. A cutoff index equal to or higher than 1.0 is considered as positive.

#### 2.2.2. Quantification of Total Anti-Spike IgG Antibodies

Humoral immune response was assessed at each time point using the Liaison^®^ SARS-CoV-2 TrimericS IgG chemiluminescent immunoassay (CLIA) on the Liaison XL analyser (Saluggia, Italy) according to the manufacturer’s instructions and using the manufacturer’s cut-off for positivity of 33.8 binding activity units (BAU)/mL. This assay quantitatively determines antibodies against the TrimericS complex, including both the Receptor Binding Domain (RBD) and N-terminal domain (NTD) sites including S1 and S2.

#### 2.2.3. Quantification of SARS-CoV-2 Neutralising Antibodies

Seroneutralisation testing (SNT) analyses were performed using a SARS-CoV-2 Wuhan-like variant (BetaCov/Belgium/Sart-Tilman/2020/1) isolated from a patient admitted at the University Hospital of Liège in March 2020 and a SARS-CoV-2 Omicron variant (Pango lineage BA.1, GISAID: EPI_ISL_7413964) obtained from an unvaccinated individual who developed moderate symptoms 11 days after returning to Belgium from Egypt. Virus isolation, expansion, titration, and SNT analysis were all performed using nonadherent sub-confluent Vero E6 cells (ATCC^®^ CRL-1586) grown in DMEM supplemented with 2% FBS and penicillin-streptomycin. The virus stocks were titrated in serial log dilutions to obtain a 50% tissue culture infective dose (TCID50) on 96-well culture plates. The plates were observed daily using an inverted optical microscope for five days to evaluate the presence of cytopathic effect (CPE), and the end-point titer was calculated according to the Reed and Muench method based on 2 × 3 replicates. Serum test samples were heat-inactivated for 40 min at 56 °C, and two-fold serial dilutions, starting from 1:10 up to 1:320, were performed in triplicate in DMEM/FBS on 96-well culture plates. Sera dilutions (50 µL/well) were then mixed with an equal volume of a pre-titrated viral solution containing 100 TCID50 of SARS-CoV-2 virus. The serum virus mixture was incubated for 1 h at 37 °C in a humidified atmosphere with 5% CO_2_. After incubation, 100 μL of a Vero cells suspension was added so that 20,000 cells were deposited in each well. The plates were then re-incubated for 5 days. For each serum, the process was repeated twice per variant by two different, trained persons. After 5 days, CPE was evaluated under light microscopy by two independent persons. Serum dilutions showing CPE were considered non-neutralising (negative), while those showing no CPE were considered neutralising (positive). Virus seroneutralisation titer was reported as the highest dilution of serum that neutralises CPE in 50% of the wells (NT_50_). If results from the 2 duplicate plates were discordant, these samples were processed again in a subsequent SNT session. For all sera showing an NT_50_ > 1:640, a second process was made using higher dilutions (up to 1:20,480). Positive (NT_50_ = 1:160, from the Belgian National Reference Centre) and negative (saline) controls were inserted in each plate.

#### 2.2.4. QuantiFERON SARS-CoV-2 Interferon-ɣ Release Assay

SARS-CoV-2-specific T-cell response was assessed in the Clinical Chemistry laboratory by a peripheral blood Interferon-Gamma-Release-immuno-Assay (IGRA) using the QuantiFERON SARS-CoV-2 research-only assay (Qiagen). The QuantiFERON^®^ SARS- CoV-2 Starter Set Blood Collection Tubes (Cat No./ID: 626115) use two Qiagen^®^ proprietary mixes of SARS-CoV-2 S-protein from the spike antigen (Ag) (S1, S2, RBD subdomains) selected to stimulate lymphocytes in heparinized whole blood samples. The QuantiFERON SARSCoV-2 Ag1 tube contains CD4^+^ epitopes derived from the S1 subunit (RBD) of the spike protein; the Ag2 tube contains both CD4^+^ and CD8^+^ epitopes from the S1 and S2 subunits of the spike protein. Briefly, one millilitre of venous blood samples was collected directly in each of the four QuantiFERON^®^ tubes containing spike peptides, positive and negative controls. The tubes were gently mixed with the whole blood to re-solubilize the content coated onto the inner walls. Whole blood was incubated at 37 °C for 16 to 24 h and centrifuged to separate plasma. IFN-γ was measured in these plasma samples using CLIA on the DiaSorin LIAISON^®^ QuantiFERON^®^-TB Gold Plus (REF:311010) and was reported in International Units per ml (IU/mL). According to the datasheet provided by the manufacturer, early data suggested an INF-ɣ cutoff for positivity at 0.15 IU/mL.

### 2.3. Statistical Analysis

Continuous variables were presented as mean and standard deviation (SD) or median and interquartile range (Q1–Q3) as appropriate. Frequency tables (numbers and percentages) were used for categorical variables. Characteristics of subjects were compared using ANOVA or Kruskal–Wallis test for continuous variables and Chi-square or Fisher exact test for categorical variables, as appropriate. Matrix of paired Spearman correlation coefficients between IFN-ɣ Ag1 and Ag2, Anti-S IgG, and neutralising antibody titers against wild type and Omicron variants were presented [12,13,14]. The strength of the positive correlation is considered as weak for absolute r values between 0.1 and 0.3, as moderate between 0.3–0.5, and as strong between 0.5 and 1. Correlations with *p*-value superior than 0.01 were considered as insignificant; in this case the correlation coefficient values are displayed in blank colour (Figure 1). The KmL3D k-means clustering method was used to build clusters of subjects with similar patterns of evolution between T0 and T1 for immune parameters [15,16,17]. Euclidean distance with Gower adjustment was used to estimate the similarity between clusters, with 100 runs for each cluster and 3 to 6 clusters [16]. The optimal number of clusters was selected by maximization of the Calinski and Harabatz criterion (Appendix A). The characteristics of the subjects in each cluster were presented and compared. Calculations used the maximum available data, and no missing values were replaced. Statistical analyses were performed using SAS (version 9.4) and R (version 4.2.2) software.

## 3. Results

### 3.1. Baseline Study Cohort Characteristics of PLWH and HCWs

A total of 119 PLWH and 79 HCWs were enrolled in the study at T0. Among them, 80 PLWH and 51 HCWs completed the whole study at T1 and constituted the study cohort. Participants’ characteristics are displayed in Table 1. Compared with HCWs, PLWH had a higher proportion of males (53.8% of PLWH versus 21.6% of HCWs, *p* < 0.001). The average age, body mass index (BMI), and proportion of comorbidities were similar between the two groups, apart from the proportion of asthma which was higher in the HCWs group (*p* < 0.01). The frequency of previous SARS-CoV-2 infection was similar between the two groups both at T0 and T1. Nine (11.2%) PLWH and six (11.7%) HCWs got infected with SARS-CoV-2 between T0 and T1, but none of these infections led to hospitalisation. A total of 69 (86.2%) PLWH and all HCWs received BNT162b2 as first two doses of vaccine. For the third dose, all HCWs and 42 (52.5%) PLWH received BNT162b2, and the remaining 38 (47.5%) PLWH received mRNA-1273. Time intervals between the first and the second vaccine dose and between the second vaccine dose and the first peripheral blood sampling (T0) were longer among PLWH compared to HCWs (*p* < 0.0001 and *p* < 0.05, respectively). In contrast, time intervals between the second and the third vaccine dose, between the third vaccine dose and the second blood sampling (T1), and between T0 and T1 were significantly shorter among PLWH when compared to HCWs (*p* < 0.0001 for each). All PLWH, except one who was infected with HIV-2, were infected with HIV-1, with a median time since diagnosis of 11 years (IQR 6.5-18). All were on antiretroviral therapy. Among PLWH, the median CD4^+^ T cell count was 743 cells/μL (IQR 592-940), and 14 PLWH (17.5%) had a CD4^+^ T cell count lower than 500 cells/µL. Only five patients (6.2%) had a viral load above 50 copies/mL.

### 3.2. Relationship between T-Cell and Humoral Immune Responses

After having characterized immune responses before and after administration of a third dose of SARS-CoV-2 vaccine among PLWH and HCWs [10], we sought to study the correlations between the different markers reflecting T-cell and humoral immune responses both at T0 and T1 in these populations. Concerning markers of T cell immune responses, we observed a strong, positive correlation in the magnitude of responses against the two different peptide pools (IFN-ɣ Ag1 and Ag2) at each time point in both groups (Spearman r at T0 = 0.89 and 0.92, Spearman r at T1 = 0.94 and 0.89, in PLWH and HCWs, respectively) (Figure 1). In the same line, concerning humoral immune response, we observed a strong, positive correlation between anti-S IgG and neutralising antibody titers against wild type (Spearman r at T0 = 0.95 and 0.87, Spearman r at T1 = 0.77 and 0.84, in PLWH and HCWs, respectively) and, to a lesser extent, against Omicron (Spearman r at T0 = 0.77 and 0.53, Spearman r at T1 = 0.74 and 0.87, in PLWH and HCWs, respectively) at each time point among both PLWH and HCWs (Figure 1). We also observed a strong, positive correlation between anti-wild-type and anti-Omicron neutralising antibodies at both time points and in both groups (Spearman r at T0 = 0.74 and 0.5, Spearman r at T1 = 0.89 and 0.9, in PLWH and HCWs, respectively). When comparing markers of T-cell and humoral immune responses, the positive correlation was less robust, though still present in both groups. We observed a moderate correlation between markers of T-cell immune response and anti-wild type neutralising antibody titers (Spearman r in PLWH at T0 = 0.4 and 0.44 for Ag1 and Ag2, Spearman r in HCWs at T0 = 0.54 and 0.53 for Ag1 and Ag2, Spearman r in PLWH at T1 = 0.47 and 0.44 for Ag1 and Ag2, Spearman r in HCWs at T1 = 0.32 and 0.37 for Ag1 and Ag2). Similarly, the correlation between markers of T cell immune response and anti-S IgG was moderate at both time points (Spearman r in PLWH at T0 = 0.39 and 0.46 for Ag1 and Ag2, Spearman r in HCWs at T0 = 0.44 and 0.44 for Ag1 and Ag2, Spearman r in PLWH at T1 = 0.37 and 0.39 for Ag1 and Ag2, Spearman r in HCWs at T1 = 0.29 and 0.4 for Ag1 and Ag2). We only found a weak positive correlation between T cell immune responses and neutralising antibody titers against the Omicron variant among HCWs at T0, while it was still moderate at T1 and among PLWH at both time points (Spearman r in HCWs at T0 = 0.17 and 0.16 for Ag1 and Ag2, Spearman r in HCWs at T1 = 0.39 and 0.44 for Ag1 and Ag2 versus Spearman r in PLWH at T0 = 0.33 and 0.38 for Ag1 and Ag2, Spearman r in PLWH at T1 = 0.31 and 0.33 for Ag1 and Ag2).

### 3.3. Cluster Analysis Identifies Four Distinct Patterns of Immune Response Evolution

We subsequently explored whether distinct patterns of T-cell and humoral immune response evolution could be observed using a non-parametric longitudinal clustering algorithm. Based on the co-evolution of markers reflecting T-cell and humoral immune responses between T0 and T1, we identified four distinct patterns of immune response evolution (Figure 2a). Participants included in the first cluster (cluster A) (71.7% of the sample, *n* = 86) demonstrated primarily, and despite a slight increase in each parameter, persistently low markers of both T-cell and humoral immune responses after the administration of the third vaccine dose (Figure 2a and Figure 3, Appendix A). Participants belonging to this cluster were mostly PLWH (62.8%, *n* = 54) including 38.9% (*n* = 21) with history of AIDS (Figure 4, Appendix A). Compared to other clusters, participants tended to have a lower BMI (25.9 ± 5.1, *p* = 0.052) (Figure 4, Appendix A). We also found that more than half of individuals (59.3%, *n* = 51) included in this cluster did not experience a SARS-CoV-2 infection in the past (*p* < 0.05) (Figure 4, Appendix A). Indeed, only 32.6% (*n* = 28) of the participants were previously infected with SARS-CoV-2 before T0 and even less (8.1%, *n* = 7) between T0 and T1. Individuals comprising the second cluster (cluster B) (17.5% of the sample, *n* = 21) were those who showed the highest median parameters reflecting T-cell immune response at both time points (1.6 and 4.4 for Ag1 and 2.2 and 6.1 for Ag2, at T0 and T1, respectively) (Figure 3, Appendix A). In the same line, participants in this cluster had the highest initial median anti-S IgG and nAbTs WT but demonstrated a less pronounced increase in these parameters between T0 and T1 compared to participants from cluster C. Median nAbTS Om was similar at both time points compared to those in cluster A. More than half of the individuals in this cluster were women (61.9 %, *n* = 13) and HCWs (61.9%, *n* = 13) (Figure 4, Appendix A). Participants in this cluster tended to be younger (42.1 ± 10.1, *p* = 0.078) compared to individuals in clusters A and C and to have a lower BMI (26.6 ± 7.4, *p* = 0.052) than individuals from cluster C. Furthermore, 52.4% (*n* = 11) experienced SARS-CoV-2 infection before T0, and only 1 participant was infected between T0 and T1. Participants in the third cluster (cluster C) (8.3% of the subsample, *n* = 10) demonstrated the lowest initial T-cell immune responses, but compared to participants in cluster A, they achieved a higher response after the administration of the third vaccine dose, although still limited. In contrast, compared to other clusters, they demonstrated the most remarkable evolution in humoral immune responses between T0 and T1, with the most robust neutralising activity against wild type and Omicron, and the highest median anti-S IgG titers at T1 (Figure 3, Appendix A). This cluster was characterised by participants who were predominantly women (90%, *n* = 9) and tended to be older (51 ± 11 years) (*p* = 0.078) and to have a higher body mass index (BMI) (31.3 ± 9.3 kg/m^2^) (*p* = 0.052) than those in cluster A and B (Figure 4, Appendix A). PLWH and HCWs represented respectively 60% (*n* = 6) and 40% (*n* = 4) of the participants in this cluster. When considering the history of SARS-CoV-2 infection, 30% (*n* = 3) had never been infected, 30% (*n* = 3) were infected before T0, and 40% (*n* = 4) were infected between T0 and T1. Thus, cluster C tended to represent older women who experienced SARS-CoV-2 with robust humoral immune responses and limited T-cell immune responses. The fourth cluster (cluster D) contained three patients with very heterogeneous characteristics and unclassifiable patterns of immune response evolution (Figure 4, Appendix A). We then focused on the SARS-CoV-2 naïve population cluster analysis which also led to the identification of four distinct patterns of immune response evolution (Figure 2b). Individuals included in the first cluster (cluster A, *n* = 35), the third cluster (cluster C, *n* = 7), and the fourth cluster (cluster, *n* = 3) showed similar patterns of immune response evolution between T0 and T1 (Figure 2b, Appendix A), compared to the cluster A, cluster B, and cluster C of the total population, respectively (Figure 2a). In contrast, the patterns of immune response evolution in the second cluster (cluster B) (*n* = 18) due to low T-cell responses at T0 and a less-pronounced increase between T0 and T1 (Figure 2b, Appendix A). We did not find any significant differences in terms of demographic and clinical factors between the different clusters (Appendix A). Concerning PLWH and HCWs sub-populations, we also identified four distinct clusters (Appendix A for PLWH and HCWs, respectively).

## 4. Discussion

Given the emergence of novel variants of concern around the world, further understanding of the characteristics and patterns of immune response evolution after SARS-CoV-2 vaccination remains important, especially for particularly vulnerable patients. Here, we explored the correlations within and between vaccine-induced immune responses before and after the administration of the third vaccine dose among PLWH and HCWs, including both previously SARS-CoV-2 uninfected and infected individuals. Based on the co-evolution of T-cell and humoral immune responses over time, we further aimed to assess the complex relationships between immune parameters and patient characteristics using longitudinal clustering.

Even if all individuals mounted detectable T-cell and humoral immune responses following vaccination, the quality of these responses was heterogeneous. We observed that T-cell and humoral immune responses appeared to be less robust in cluster A compared to clusters B and C. This cluster included mainly, for more than half of its individuals, PLWH who had never been infected with SARS-CoV-2. We recently demonstrated that SARS-CoV-2-specific IFN-ɣ production after third dose was significantly lower precisely among those SARS-CoV-2 naïve PLWH when compared with HCWs, raising concerns about the vaccine’s ability to induce protective T-cell immune response among PLWH who had not been previously infected [10]. This is especially concerning since vaccine-induced cross-reactivity against Omicron seems to rely mainly on T-cell response, contrasting sharply with the markedly low Omicron-specific antibody response [7].

Cluster C, whose individuals showed a particularly drastic increase in markers of humoral immune response following the third dose of vaccine, was mainly comprised of female participants who experienced SARS-CoV-2. Previous studies demonstrated that, although SARS-CoV-2 infection tended to confer a more sustained protection against the virus than vaccination, this protection can wane and, therefore, should not detract from the need for vaccination [18]. Indeed, the immunity conferred by the combination of vaccination and previous infection, called hybrid immunity, confers the highest magnitude and durability of protection against reinfection, hospital admission, or severe disease [18,19,20]. This concept is further supported by our observations. Moreover, it is well known that females generally respond differently than males to many vaccines, which may be due in part to their heightened immune response, and this trend seems also true for the novel SARS-CoV-2 vaccines [21].

Lastly, we evaluated how well T-cell and humoral immune responses correlated with each other. While we observed a strong positive correlation between markers mirroring humoral immune response, markers of T-cell response following vaccination correlated only in a lesser extent with markers of humoral immunity. This suggests that neutralising antibody titers are not always a reliable reflection of the magnitude of the immune response as a whole and should be interpreted with caution when used alone to assess vaccines’ immunogenicity [17,22].

The Higher Health Council in Belgium recommends an additional booster (in addition to the basic vaccination and the first booster) of the vaccination against SARS-CoV-2 for groups at-risk as PLWH [23]. Vaccine acceptance was already a global concern prior to the COVID-19 pandemic [24]. In the Belgian population, 62.5% had received a third dose of the vaccine and even less a fourth or a fifth dose (33.5 and 4.3%, respectively) [25,26], like many of our neighbouring countries [27]. Vaccination hesitancy does not spare at-risk populations, such as PLWH [27,28]. The increased risk of severe COVID-19 makes vaccination a priority for PLWH. However, barriers to vaccine acceptance including concerns about vaccine risk, efficacy, and safety have been reported [29]. Furthermore, COVID-19 recommendations in PLWH were limited to people with a CD4^+^ T cell count less than 200 cells/µL [23]. Nonetheless, our study demonstrated that SARS-CoV-2-specific T-cell immune responses after the third dose was significantly lower, precisely among those SARS-CoV-2 naïve PLWH, regardless of CD4^+^ T cell count [10,30,31,32,33]. Future research assessing vaccine-induced immune responses among PLWH are necessary to determine who should benefit the most from supplementary vaccination and subsequently which subgroup we need to convince first to promote uptake of SARS-CoV-2 vaccination. Identifying clinical and biological markers to recognize this subgroup could be useful. It is vital from a public health perspective to address the PLWH subgroups the most at risk for COVID-19, increase confidence in the various institutions and newly developed vaccines, establish a multisectoral approach, and implement campaigns to debunk misinformation [34,35].

Our study presents some limitations. First, a greater number of subjects is needed to draw more solid conclusions about the efficiency of the vaccine-induced immune response against SARS-CoV-2 in PLWH, given the heterogeneity in matter of depth and characteristics of immunodeficiency in this population. Second, we did not test which variant caused the infection in participants who experienced SARS-CoV-2, which may be relevant when studying subsequent vaccine responses. Third, our analysis did not investigate the timing between vaccination and previous infection. Furthermore, we did not distinguish between asymptomatic and symptomatic infections, which may induce different immune responses, although we do know that no participant was hospitalised for SARS-CoV-2 infection, suggesting that the infections were either asymptomatic or mild. Another relevant consideration is that recruitment periods and intervals between time points were not precisely the same between PLWH and HCWs. Indeed, sampling at T0 had been performed earlier for HCWs, before the emergence of Omicron, preventing our HIV-negative population from being infected by this specific variant. However, in counterpart, to overcome possible resulting bias, all analyses were adjusted for the parameters that had a significant impact on at least one variable of interest. Finally, the main limitation of K-means clustering is to determine the exact number of clusters reflecting clinically meaningful differences. Here, we used the maximization of the Calinski and Harabatz criterion as an indication of the optimal number of clusters. Interestingly, we identified the same number of clusters in the entire cohort, SARS-CoV-2 naïve individuals, SARS-CoV-2 naïve PLWH, and SARS-CoV-2 naïve HCWs.

## 5. Conclusions

In this study, we provide important insights into the dynamics and heterogeneity of T-cell and humoral immune responses among SARS-CoV-2-vaccinated people living with HIV and HIV-negative participants. We identified three distinct patterns of immune response evolution which were representative of clusters of individuals with distinct immune features. While T-cell and humoral responses correlate in some individuals, their discordance in others highlights the complex interactions of the immune system among vaccinated individuals and indicates that there are several mechanisms by which protection against SARS-CoV-2 can be achieved. Prior or breakthrough natural infection can enhance the activity of vaccines and must be taken into account for informing global vaccine strategies among PLWH, even with a viro-immunologically controlled infection.

## Figures and Tables

**Figure 1 viruses-15-01435-f001:**
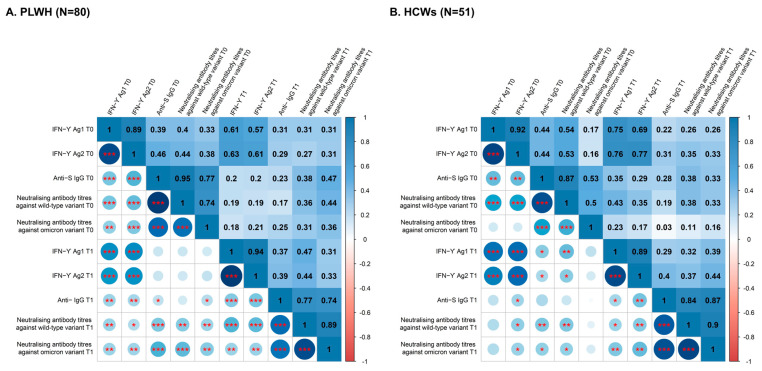
Spearman correlation matrix between IFN−ɣ Ag1, IFN−ɣ Ag2, Anti-S IgG, neutralising antibody titers against wild type, and neutralising antibody titers against Omicron variants at T0 and T1 among PLWH (**A**) and HCWs (**B**). Positive correlations are displayed in blue, and negative correlations are displayed in red. Colour intensity is proportional to the correlation coefficients. Correlations with *p*-value > 0.01 were considered as insignificant; in this case, the correlation coefficient values are displayed in blank colour. *p*-value: *** < 0.001, ** < 0.01, * < 0.05.

**Figure 2 viruses-15-01435-f002:**
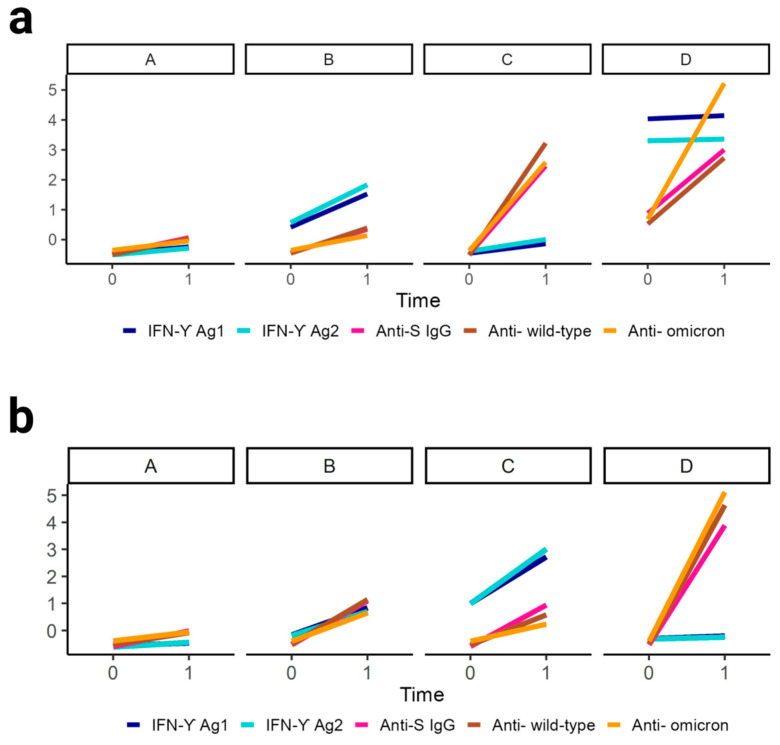
Clustering of T-cell and humoral immune responses evolution. IFN-ɣ Ag1, IFN-ɣ Ag2, Anti-S IgG, and neutralising antibody titers against wild type and Omicron variants patterns of evolution in each of the four clusters (**a**) among all individuals either SARS-CoV-2 experienced and naïve (total n = 117; cluster A: n = 86, cluster B: n = 21, cluster C: n = 10, cluster D: n = 3), (**b**) among SARS-CoV-2 naïve individuals (cluster A: n = 35, cluster B: n = 18, cluster C: n = 7, cluster D: n = 3).

**Figure 3 viruses-15-01435-f003:**
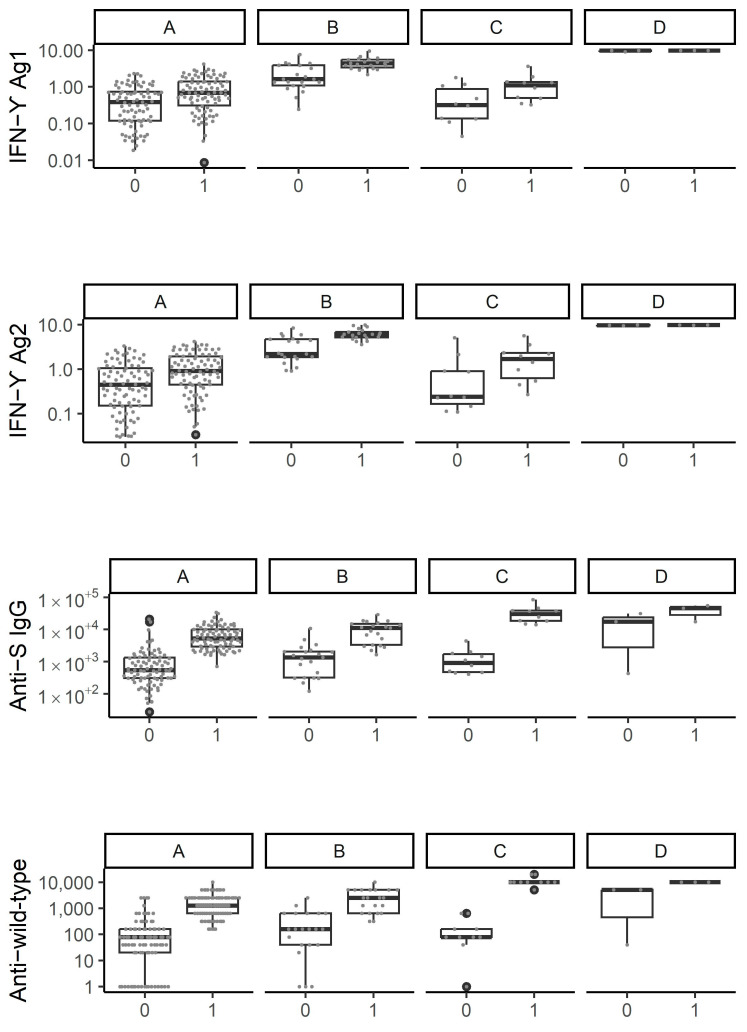
IFN-ɣ Ag1 and Ag2, Anti-S IgG, and neutralizing antibody titers against wild type and Omicron variants in each of the four clusters among all individuals either SARS-CoV-2 experienced and naïve (total n = 117; cluster A: n = 86, cluster B: n = 21, cluster C: n = 10, cluster D: n = 3), among SARS-CoV-2 naïve individuals (cluster A: n = 35, cluster B: n = 18, cluster C: n = 7, cluster D: n = 3).

**Figure 4 viruses-15-01435-f004:**
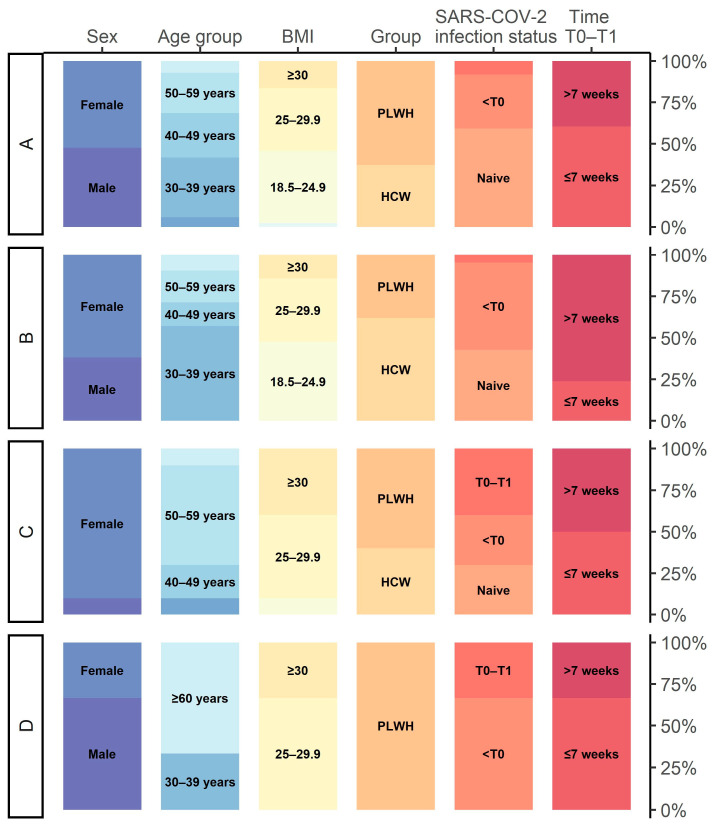
Background characteristics of individuals in each of the four clusters among all individuals either SARS-CoV-2 experienced and naïve (total n = 117; cluster A: n = 86, cluster B: n = 21, cluster C: n = 10, cluster D: n = 3), among SARS-CoV-2 naïve individuals (cluster A: n = 35, cluster B: n = 18, cluster C: n = 7, cluster D: n = 3).

**Table 1 viruses-15-01435-t001:** Background characteristics of PLWH and HCWs individuals.

Variable	All (n = 131)	PLWH (n = 80)	HCWs (n = 51)	*p*-Value
Gender				
Male	54 (41.2)	43 (53.8)	11 (21.6)	0.0003
Age (Years)	44.6 ± 10.5	45.6 ± 10.7	43.0 ± 10.0	0.18
18–29	6 (4.6)	4 (5.0)	2 (3.9)	
30–39	46 (35.1)	24 (30.0)	22 (43.1)	
40–49	34 (26.0)	21 (26.2)	13 (25.5)	
50–59	32 (24.4)	22 (27.5)	10 (19.6)	
≥60	13 (9.9)	9 (11.3)	4 (7.8)	
BMI (kg/m^2^)	26.9 ± 6.1, n = 130	27.5 ± 5.6	25.9 ± 6.9, n = 50	0.13
Underweight (<18.5)	2 (1.5)	0 (0.0)	2 (4.0)	
Normal range (18.5–24.9)	51 (39.2)	29 (36.2)	22 (44.0)	
Overweight (25–29.9)	51 (39.2)	34 (42.5)	17 (34.0)	
Obese (≥30)	26 (20.0)	17 (21.3)	9 (18.0)	
Ethnicity				-
Caucasian	-	34 (42.5)	-	
African	-	41 (51.3)	-	
Other	-	5 (6.2)	-	
Medical history				
Diabetes mellitus	6 (4.6)	5 (6.2)	1 (2.0)	0.40
Hypertension	25 (19.1)	18 (22.5)	7 (13.7)	0.21
Heart failure coronary artery disease	2 (1.5)	2 (2.5)	0 (0.0)	-
Stroke	1 (0.8)	1 (1.2)	0 (0.0)	-
Liver disease	1 (0.8)	1 (1.2)	0 (0.0)	-
Kidney disease	0 (0.0)	0 (0.0)	0 (0.0)	-
Chronic lung disease	1 (0.8)	1 (1.2)	0 (0.0)	-
Asthma	3 (2.3)	0 (0.0)	3 (5.9)	0.0028
Autoimmune disease	2 (1.5)	0 (0.0)	2 (3.9)	-
Hematological cancer	1 (0.8)	0 (0.0)	1 (2.0)	-
Non hematological cancer	11 (8.4)	7 (8.8)	4 (7.8)	1.0
Solid-organ/cell transplantation	0 (0.0)	0 (0.0)	0 (0.0)	-
Immunosuppressive drugs				-
Corticosteroids	0 (0.0)	0 (0.0)	0 (0.0)	
Other	0 (0.0)	0 (0.0)	0 (0.0)	
Previous SARS-CoV-2 infection (before T0)				
Questionnaire	29 (22.1)	14 (17.5)	15 (29.4)	0.11
Positive anti-N antibody	40 (30.8), n = 131	30 (37.5)	10 (20.0), n = 50	0.035
SARS-CoV-2 experienced *	48 (36.6)	32 (40.0)	16 (31.4)	0.32
Previous SARS-CoV-2 infection (before T1)				
Questionnaire	33 (25.2)	15 (18.8)	18 (35.3)	0.033
Positive anti-N antibody	57 (43.9), n = 130	40 (50.0)	17 (34.0), n = 50	0.074
SARS-CoV-2 experienced *	63 (48.1)	41 (51.2)	22 (43.1)	0.37
Experienced (between T0 and T1)	15 (11.4)	9 (11.2)	6 (11.7)	-
First vaccine dose				-
BNT162b2 mRNA (Pfizer)	120 (91.6)	69 (86.2)	51 (100.0)	
mRNA-1273 (Moderna)	4 (3.0)	4 (5.0)	0 (0.0)	
ChAdOx1-S (Astra Zeneca)	7 (5.4)	7 (8.8)	0 (0.0)	
Second vaccine dose				-
BNT162b2 mRNA (Pfizer)	120 (91.6)	69 (86.2)	51 (100.0)	
mRNA-1273 (Moderna)	4 (3.0)	4 (5.0)	0 (0.0)	
ChAdOx1-S (Astra Zeneca)	7 (5.4)	7 (8.8)	0 (0.0)	
Third vaccine dose				-
BNT162b2 mRNA (Pfizer)	93 (71.0)	42 (52.5)	51 (100.0)	
mRNA-1273 (Moderna)	38 (29.0)	38 (47.5)	0 (0.0)	
Time between first and second vaccine dose (weeks)	4.0 (3.0–5.0)	5.0 (4.4–5.0)	3.0 (3.0–3.1)	<0.0001
Time between second vaccine dose and sample at T0 (weeks)	24 (24–26)	25 (23–27)	24 (24–24)	0.014
Time between second and third vaccine dose (weeks)	32 (26–38)	27 (25-31)	38 (35–39)	<0.0001
Time between third vaccine dose and sample at T1 (weeks)	3.7 (2.9–4.7)	2.4 (3.1–3.9)	4.7 (4.0–8.0)	<0.0001
Time between T0 and T1 (weeks)	7 (4–19)	5 (4–6)	19 (18–19)	<0.0001
HIV infection				-
HIV-1	-	79 (98.8)	-	
HIV-2	-	1 (1.2)	-	
Time at T0 since HIV diagnosis (years)		11 (6.5–18)		-
<1	-	1 (1.2)	-	
1–5	-	17 (21.3)	-	
6–10	-	17 (21.3)	-	
>10	-	45 (56.2)	-	
Nadir CD4^+^ T cell count per μL	-	292 (166–502)	-	
<200	-	25 (31.2)	-	
≥200	-	55 (68.8)	-	
Last CD4^+^ T cell count per μL (2021 or 2022)		743 (592–940)		-
<350	-	3 (3.7)	-	
350–499	-	11 (13.8)	-	
≥500	-	66 (82.5)	-	
CD4/CD8 ratio, n = 117		1.1 ± 0.57		
0.6–1	-	26 (32.5)	-	
>1	-	38 (47.5)	-	
Last plasma viral load copies/mL		<20 (<20–<20)		-
<50	-	75 (93.8)	-	
≥50	-	5 (6.2)	-	
ART regimen				
Dual therapy	-	26 (32.5)	-	
NRTI + INI	-	22 (27.5)	-	
NNRTI + INI	-	4 (5.0)	-	
>2 ART	-	54 (67.5)	-	
NRTI + NRTI + INI	-	35 (43.8)	-	
NRTI + NRTI + NNRTI	-	13 (16.2)	-	
NRTI + NRTI + PI	-	2 (2.5)	-	
NRTI + NRTI + PI + INI	-	2 (2.5)	-	
NRTI + INI + PI	-	1 (1.2)	-	
MVC + NRTI + INI + PI	-	1 (1.2)	-	
Time on ART (years)	-	10.7 ± 6.9	-	
CMV IgG positive	-	70 (97.2), n = 72	-	
HBsAg positive	-	1 (1.2)	-	
HCV-Ab positive	-	3 (3.8)	-	
Influenza vaccine the same year	-	33 (25.2)	-	

Results are expressed as n (%), mean ± SD, or Median (Q1–Q3) as appropriate and *p*-values of Chi-square or Fisher exact test, ANOVA, or Kruskal–Wallis test, respectively. *: Yes if positive through questionnaire or anti-nucleocapsid antibodies.

## Data Availability

The data presented in this study are available on request from the corresponding author. The data are not publicly available due to restrictions on privacy.

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
