# Peer review of "Cluster Analysis Identifies Distinct Patterns of T-Cell and Humoral Immune Responses Evolution Following a Third Dose of SARS-CoV-2 Vaccine in People Living with HIV"

_viruses, 2023, doi:10.3390/v15071435_

Round 1

Reviewer 1 Report

A few minor comments for authors' consideration:

RESULTS:

-lines 203-208: please relocate to DISCUSSION

-please present the clustering model diagnostic summary (Calinski and Harabatz criterion) comparison table and include in Supplementary file

-the presented information is partly duplicated between Figure 2a and Table 3, Figure 2c and Table 2. Highly recommended to simplify Table 2, and you may add Table 3 data to Figure 2a, and then relocate Table 3 to supplementary. 

DISCUSSION

-suggest to add 1-2 sentences about the strength or implication of cluster analysis

-not sure if "trajectory" is a suitable term as just 2 time points were involved. 

Reviewer 2 Report

El Moussaoui et al. present here a follow-up study of their previous J Infect. 2022 story. In this previous study, they found lower immune responses in people living with HIV than in uninfected people. This observation is important because it may mean that people under ART may be less susceptible to SARS-CoV-2 vaccination and consequently remain a population at risk. Here, they found associations between T cell responses and humoral responses in PLWH and uninfected people, after vaccination against SARS-CoV-2. They identified 4 immune trajectories : 1) low responders, mostly PLWH, 2) good responses, mostly young participants, 3) strong B cell responses, but weak T cell responses, mostly older woman, and 4) heterogenous non-classified trajectories, suggesting that people in these different group of people would benefit differentially from SARS-CoV-2 vaccination.

The paper provide an interesting insight of SARS-CoV-2 vaccination. I appreciate the effort to study PLWH, a group of people than can apparently appear at risk. I do have concern in term of clarity, novelty in regards of their previous paper, and interpretation of the data, all concerns I expect could be mitigated after a careful revision

11) Knowledge of their previous work is clearly a prerequisite. However, it makes it difficult to understand the results in the current manuscript. For example, the magnitudes of T and Ab responses are present. It is odd to present correlation and trajectories without presenting magnitudes first.

22) To my understanding, the authors cannot discriminate between CD4 and CD8 T cell responses. This is a serious flaw, as I do not expect CD8 to necessarily correlate with humoral responses. Their correlation may be underestimated for that reason.

T 3)  The authors should try temporal correlation between T0 and T1. Previous reports showed better correlations that way (PMID: 35732172, PMID: 34648302, PMID: 34453880 ). There references should be added to the manuscript.

44) Pre-exposure to SARS-CoV-2 was reported to dramatically change immune trajectories. Authors should focus on SARS-CoV-2 naïve individual, and consider participants with pre-exposure in distinct groups

5 5) In correlation plots, the authors should not hide the r values when the p is not reaching significance. This is misleading, as the threshold of p < 0.01 is arbitrary. Instead, add asterisk for r values that reach significance.

6 6) I was confused by figure 2, I do not find it conveys a clear message. From their previous work, we already expect that PLWH will have a different trajectory. Why not directly compare between PLWH and infected?

77) The timing labels are very confusing. T0 and T1 are widely used to indicate baseline and dose 1, respectively. I would recommend to change this to something less confusing.
